# Cash Transfer Programmes in Pakistan through a Child Well-Being Lens

**Altaf Hussain \* and Susanne Schech** 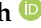

College of Humanities, Arts and Social Sciences, Flinders University, Adelaide, SA 5001, Australia;
susanne.schech@flinders.edu.au
\* Correspondence: altaf.hussain@flinders.edu.au

**Abstract:** This paper analyses data from a qualitative study undertaken with children and their families in two cash transfer programmes (CTPs) in Pakistan. Using a three-dimensional child well-being model that distinguishes material, relational and subjective dimensions, it argues that CTPs have helped extremely poor families sustain their basic dietary needs and marginally increase their health spending. Additional conditional payments have led to increased primary school enrolments, but CTPs have failed to address the distinctive vulnerabilities of children, including their nutritional needs, relational well-being and social status. A more holistic and child-sensitive approach to social protection would be the way forward to improve child well-being in line with the United Nations Charter on Rights of Children (UNCRC) to which Pakistan is a signatory.

**Keywords:** cash transfer programmes; well-being; poverty; children; Pakistan





## 1. Introduction

Since first launched in Mexico and Brazil in the late 1990s, cash transfer programmes (CTPs) have rapidly spread across the world and now play a prominent role in government social protection and anti-poverty strategies (Rutherford and Bachay 2017; Peck and Theodore 2015). Many CTPs are selective and target specific groups, such as extremely poor households, and some also make cash payments conditional on specific actions by the recipients (Eskelinen and Perkiö 2018). Conditional cash transfer programmes (CCTPs) target individuals classified as 'poor' who in turn are required to fulfil certain conditions, such as participate in immunization programmes, stop sending their children to work, or enrol their children at school (Ballard 2012). A large body of research on the multiple dimensions of poverty and diverse experiences of poverty supports calls to make CTPs more sensitive to the needs and interests of specific groups, including women and children (Esser et al. 2019). Existing cash transfer mechanisms have proven useful in mitigating the immediate economic impacts of the COVID-19 pandemic. However, according to Sumner et al. (2020), COVID-19 will have serious longer-term consequences for the world with the proportion of the global population living in extreme poverty increasing by an estimated half a billion, and children's well-being likely to suffer more compared to adults. School closures due to COVID-19's impact on children's education, mental health and nutrition expose them to family violence and child-labour and potentially lock them in a generational spiral of poverty (OECD 2020; Thevenon and Adema 2020).

Children are a group that requires special consideration in poverty reduction interventions because their vulnerabilities to, and experiences of, poverty are different from those of adults (Alkire and Roche 2012). Sabates-Wheeler and Roelen (2011) have identified three types of child-specific vulnerabilities that relate to children's specific biological and physio-social needs, dependency on adults, and institutionalised disadvantage. Firstly, children are more vulnerable to suffering long-lasting detrimental consequences for their development—and the country's human capital—when their nutrition, health care and educational needs are not met. Secondly, children depend on adults for ensuring that their

needs are met, as well as for care and protection, and these relational power differences render children vulnerable to verbal, physical or sexual abuse by adults. Thirdly, due to the low value societies accord to children's views and rights, they are invisible and unrepresented in public policy debates and unable to exercise their voice. As Sabates-Wheeler and Roelen (2011, p. 186) point out, this vulnerability "can be considered a social and cultural artefact that is put in place and reinforced by institutional structures", and addressing it requires deeper structural change.

With children's unique needs and rights becoming more visible on the international development agenda, spearheaded by global initiatives such as the United Nations Convention on Rights of Child (UN General Assembly 1989) and the Sustainable Development Goals (UNICEF 2019), the pressure on CTPs to become more child-sensitive has grown. A three-dimensional well-being model developed by Andy Sumner (2010) offers a useful framework for examining how CTPs impact on children's vulnerabilities and well-being. In addition to children's education, health and nutrition, which are commonly targeted by CTPs, Sumner's model also addresses the less tangible aspects of well-being such as social inclusion, protection from violence and abuse, social and family roles, agency, autonomy, status and fun and play. This paper explores the impacts of CTPs in Pakistan on these multiple dimensions of well-being of children by including their own perspectives alongside household-level data.

A significant majority of Pakistan's estimated 80 million children under 18 years of age lack basic amenities of life including water, shelter, food, education and health. Child education, as the bedrock of the progress and growth of societies, is at its lowest ebb, with the country failing to achieve the education enrolment target set by United Nations Millennium Development Goals (MDGs) by a long margin. Estimates that "over 22.6 million children aged 5–16 are out of school, including 5 million of primary school age" and evidence of drastic variations in school attendance in different regions of the country (UNICEF 2016) paint a grim picture of children's future livelihoods. Malnutrition is chronic and stretches over generations, as indicated by 44 percent of Pakistani children suffering from stunting and 54 percent of children under 5 being deficient in Vitamin A (UNICEF 2016). The Society for the Protection of the Rights of the Child (SPARC 2018) has reported widespread violence against children, including physical punishment in schools, murder, rape, sodomy, torture, trafficking and police torture. Child sexual abuse is increasing, reaching an unprecedented high level in 2018–2019, and Pakistan's children are particularly vulnerable to child and bonded labour with the country ranked 8 among 167 countries in the Global Slavery Index (2018). At the same time, however, the government of Pakistan has increased its efforts to address poverty, and the country's poverty rate has dropped significantly from 50.4 percent in 2005–2006 to 24.3 percent in 2018 (Government of Pakistan 2019). CTPs play a pivotal role in the country's poverty reduction strategy, but there has been limited attention on their impact on children's well-being. This paper aims to extend the discussion on the impacts of CTPs on well-being in Pakistan through a qualitative case study that includes children's own perspectives alongside household-level data. This is a qualitative study undertaken with beneficiaries of two programmes—Pakistan Bait-ul-Mal (PBM) and Benazir Income Support Programme (BISP). The first author conducted focus group discussions with children aged 10 years and older to gain their perspectives on subjective and relational well-being, including their concept of a good life, satisfaction and other dimensions of well-being identified in Sumner's (2010) model. This qualitative data is contextualized through a socioeconomic survey of 50 households and interviews with the primary recipients of the cash payments to gauge the level of material well-being of children achieved through these programmes.

Before presenting the research methods and findings, the section below provides a brief analysis of the existing literature on CTPs and their impacts on child poverty and child well-being.

**2. Cash Transfer Programmes and Child Well-Being: Review of Literature**

One important reason why CTPs have become so popular is that they have produced measurable evidence of progress in key areas concerning child well-being, as argued by numerous evaluation studies. According to research from around the world, the main ways in which children benefited from CTPs is through increased access to schooling, improved health and nutrition and a reduction in the need for children to work to supplement household income. As child-sensitive CTPs focus on education and health to address children's needs, they are more effective when they enjoy the commitment of the highest level of government, which has to invest in public education and health services so that poor families can access them. One example is South Africa, where the government has increased expenditure on education to 6.2% of the GDP in 2018 (World Bank n.d.) and boosted school enrolment by financially supporting poor families through a means tested unconditional Child Support Grants programme (Adato et al. 2016). CTPs have been reported to increase school enrolments in many other countries, including in Zambia, where 10.4 percent more 5- and 6-year-old children enrolled under the Social Cash Transfer Scheme (SCTS), and in Malawi, where the Mchinji Cash Transfer programme reportedly has doubled new school enrolments (Adato and Bassett 2009; UNICEF 2015). Brazil's Bolsa Escola programme, Ecuador's Bono de Desarrollo programme, Turkey's cash transfer programme, Bangladesh's Reaching Out of School Children Programme (ROSC) and Pakistan' Benazir Income Support Programme (BISP) all reported significant impacts on children's schooling, particularly when cash payments were made conditional on maintaining school attendance (Adato and Bassett 2009; Cheema et al. 2016).

School and infant feeding programmes contributed to these achievements. One school programme in Kenya demonstrated significant improvements in protein intake in 80 percent of the students and vitamin A intake in about 20 percent (UNICEF 2015). Programmes for infants such as Programa de Abasto Social de Leche in Mexico have helped in achieving improved weight-for-age for pre-school children to 60 months, better height and weight measurement levels and higher food intake spending (Sanfilippo et al. 2012). CTPs have also aimed at improving the health of children more broadly through a variety of interventions. For example, Nicaragua's RPS programme helped increase visits to hospitals by 16.3 percent by families with children under three years of age. Results from Mexico's PROGRESA programme demonstrate an increase in perinatal visits in the early stages of pregnancy and clinical growth monitoring visits by beneficiaries under the age of two years, as well as decreasing illness rates among children under five, as reported by mothers (Adato and Bassett 2009). There is some evidence of the extent to which CTPs have helped address psychosocial vulnerabilities and mental health issues of children. Abu-Hamad et al.'s study in Gaza found that cash transfers reduced stress through relieving economic hardship in poor households but did not tackle important problems commonly experienced by children in Gaza, such as inadequate care and physical and psychosocial violence (Abu-Hamad et al. 2014, p. 122). In Kenya, mental health impacts were also limited despite the CTP being targeted to young people: the Cash Transfer Program for Orphans and Vulnerable Children achieved a reduction in depression as a result of receiving cash transfers that was only significant among young men aged 20–24 years (Kilburn et al. 2016). A recent review of literature on the impacts of cash transfers on mental health in children and young people deduces that such programmes could have positive impacts on their mental health but did not find a direct connection between cash transfers and depression (Zimmerman et al. 2021).

Some CTPs address aspects of relational well-being, which refers to relations of love and care, networks of support and obligation, relations with the state, social, political and cultural identities and the scope for personal and collective action and influence. In South Africa and Brazil, CTPs conditioned parents to stay in their regions/localities to qualify for support instead of moving to cities for employment, which helped to keep families together and encouraged fathers to spend time with their children. Similarly, in Colombia, Families in Action and other programmes were designed to reunite families

and stop the forced migration of parents due to 'social conflicts' and lack of employment (Barrientos et al. 2013).

Recent studies indicate that BISP's unconditional transfer has a positive impact on the school enrolment of children (Churchill et al. 2021; Afzal et al. 2019). However, no impact on child labour could be demonstrated in the short-term. This may be due to beneficiary households expanding their agricultural activity significantly, as found in other studies elsewhere. For example, De Hoop et al. (2020) found that cash transfers in Zambia and Malawi created increased labour demands on farmlands which led to children being more involved in farming and livestock care.

Despite mounting evidence that CTPs can contribute to children's well-being in various ways, limitations have also been identified which relate to gaps in information and data collection, assumptions about intra-household distribution of benefits, and the lack of attention to children's non-material needs and interests (Sabates-Wheeler and Roelen 2011; Main 2014; Redmond 2008). In relation to the first limitation, significant data gaps have been identified by UNICEF, as the global custodian of data for children, stating that most countries either have insufficient data or show insufficient progress to meet global SDG targets by 2030 in the 44 indicators that directly concern children (UNICEF 2019). Even in the world's richer countries, data coverage for children and young people is severely deficient for Goals 1 (poverty), 5 (gender equality), 11 (sustainable cities and communities) and 16 (peace, justice and strong institutions), according to a study prepared for the OECD (Marguerit et al. 2018, p. 5). The lack of child-sensitive data affects cash transfer programme design and evaluation. For example, the BISP in Pakistan defined children's vulnerabilities narrowly in terms of nutrition, immunization and school enrolment (Cheema et al. 2016), which are more readily measured than relational aspects of well-being.

The second limitation relates to the tendency in CTPs to conceptualise the problem of poverty as one that can be addressed by increasing household spending on basic needs. There is a simplistic theory of change at work which assumes that material incentives and conditionalities lead to decisions and behaviours that improve well-being (Adato et al. 2016; Cheema et al. 2016). An associated assumption is that cash benefits will be distributed in ways from which children benefit. However, evidence of inter-generational conflict over access to cash transfers and competing spending priorities suggests that the complex and context-specific causes of poverty, and the relationships between carers and children, require more attention in the design and delivery of CTPs (Sabates-Wheeler and Roelen 2011). Frequently, "children's voicelessness and lack of autonomy limit their ability to counteract" decisions made by their carers that go against children's strategic needs, and thus, "CCTs can be considered to perpetuate and reinforce children's dependent and marginalised position" (Sabates-Wheeler and Roelen 2011, p. 190). The importance of relational vulnerabilities connects to the third limitation of CTPs that most either ignore children's non-material needs and interests or consider them only indirectly. Taking into account less-tangible needs requires gathering information from children about their own perspectives on their situation, which few studies and evaluations have done. Redmond's (2008) review of qualitative research on children's perspectives on poverty in richer countries highlights three key findings: firstly, social exclusion hurts children more than material deprivation itself; secondly, family relations often protect children but can also hurt them, as economic pressures cause arguments and sometimes violence in the family; and thirdly, children's agency in response to poverty tends to focus at the everyday personal level, such as earning money, helping parents with domestic work and child care and improving their prospects through education. Similar findings emerged from Witter and Bukokhe's (2004) study in Uganda where children had a broad and rich understanding of poverty, placed emphasis on personal and family factors and were eager to participate in mitigating poverty. More recently, there has been increased interest in qualitative, locally grounded research to assess the social impacts of CTPs (Molyneux et al. 2016), and this paper contributes a child-focused case study in Shaheed Benazirbad district in Pakistan's Sindh province.

## 3. Conceptual Framework and Research Methods

To promote the inclusion of children's perspectives, needs and interests into poverty reduction interventions, Sumner's (2010) well-being model is a useful lens on the distinctive aspects of child poverty and well-being. Drawing on Amartya Sen's (2005) work on capabilities, it shifts the focus from income and human development indices to what children can be and can do. According to Sumner, conventional approaches to wellbeing contemplate mostly the material aspects including education, health, food, and living conditions. In this context, his multi-dimensional approach to well-being takes into account the comprehensive rights of children as enshrined in United Nations Convention on Rights of Child (UNCRC), and specifically, material, relational and subjective well-being (Table 1).

**Table 1.** Characteristics of well-being model.

| Determinants of Well-Being | Material | Relational | Subjective |
|---|---|---|---|
| Indicators | Needs satisfaction indicators Material asset indicators | Human agency indicators Multi-dimensional resource indicators | Quality of life indicators |
| Key determinants | Income, wealth and assets Employment and livelihood activities Education and skills Physical health and (dis)ability Access to services and amenities Environmental quality | Relations of love and care Networks of support and obligation Relations with state, law, politics, welfare, social, political and cultural identities and inequalities Violence, conflict and (in)security Scope for personal and collective action and influence | Understandings of sacred and moral order Self-concept and personality Hopes, fears and aspirations Sense of meaning/meaninglessness Levels of (dis)satisfaction Trust and confidence |

Source: Sumner (Sumner 2010, p. 1067).

The relational and subjective dimensions of well-being in Sumner's model bring into focus the cultural and social factors in children's poverty and the crucial role of relationships and status in their perceptions and experiences of poverty (Witter and Bukokhe 2004). In addition to the material well-being which many CCTs address, the model also attends to family relations and networks of support, political, social and cultural relations, security and agency and subjective views of well-being which include hopes, fears, trust and aspirations. These dimensions should be part of an all-encompassing approach to well-being capable of addressing risks that can lead to a break-down of family care systems and leave children exposed to external care from state or charitable institutions (Sumner 2010).

### 3.1. Pakistan's Cash Transfer Programmes: Pakistan Bait-ul-Mal and Benazir Income Support Programme

This framework informs our analysis of two Pakistani CTPs. The first, Pakistan Bait-ul-Mal (PBM) aims to provide financial support to the "poorest of poor" families and disrupt the transmission of intergenerational poverty by helping them enrol their children in schools and build human capital (Sayeed 2015). PBM was launched after International Labour Organization's (ILO) survey in 1996 revealed Pakistan had about 3.3 million children in hazardous child labour conditions (PBM 2018). Families of PBM beneficiary children between 5–16 years receive PKR 300 per month subject to their enrolments in schools and an additional PKR 10 are given to children as stipend through parents' accounts. This programme uses the Benazir Income Support Programme's poverty score card database to reach eligible children.

The second programme, Benazir Income Support Programme (BISP), was launched in 2008 to address the rising food and fuel prices in the country in the wake of the global financial crisis (Gazdar and Zuberi 2014). This unconditional cash transfer programme focuses on families and identifies as its primary recipients "poor women, with an immediate

objective of consumption smoothing and cushioning the negative effects of slow economic growth", and by 2016, 5.7 million families were receiving cash payments of PKR 18,000 per year under this programme (equivalent to USD 171 in 2016) (BISP 2018).

*3.2. Frequency and Value of Transfers*

At the start of the programme in 2008, the value of annual cash transfers made to beneficiary households under the BISP programme was PKR 1000 per month. It soon increased to PKR 1200 in 2012 and PKR 1500 in 2014 (BISP 2018). This was the level of payments when the research for this paper was conducted in 2015. These transfers were made quarterly, and beneficiary households received the annual sum of PKR 18,000 per year in 4 tranches (Cheema et al. 2016).

In 2012, a conditional cash transfer program, Waseela-e-Taleem, was added to BISP to financially support the primary education of 5–12-year-old children of BISP beneficiary families. Eligible BISP recipient households receive an additional payment of PKR 750 per child every quarter if they maintained a 70 percent school attendance rate. In combination, these programmes are intended to contribute towards achieving Sustainable Development Goals (SDGs) 1, 4 and 5 by addressing poverty and lack of education. While BISP has been extensively evaluated (Cheema et al. 2016), there has to date been little attention paid to children's well-being.

*3.3. Research Methods*

The present case study relies on a combination of research methods including a survey of 50 households, 20 semi-structured interviews with mothers in beneficiary families and 3 group discussions with children. Although BISP and PBM have country-wide coverage, the research focuses on Shaheed Benazirabad district of Sindh where both programmes are rolled out. Fieldwork was conducted in July and August 2015 by the first author who is fluent in Urdu and Sindhi, the local language. The research was conducted as part of a higher degree and approved through the University research ethics process in the Netherlands, and in addition, local ethical considerations in Pakistan were taken into account. This involved following a locally accepted approach of seeking permission from community elders for the research to take place, and then approaching female parents, the recipients of CTs, in beneficiary households identified through purposive sampling with the help of a local non-government organization (NGO) working with beneficiary households of BSIP and PBM in the area. Women who agreed to participate contributed to the survey and answered additional interview questions. The survey asked questions about age and education of household members, household income including cash transfer, and how it was used for food, health, education and paying back loans. This data was cross tabulated to generate descriptive information about household well-being. For the semi-structured interview component, an open coding method was adopted where the analysis focused on income security, benefits for children, child agency, girls' education, child labour and good life.

For the focus groups with children, parents' permission was sought for their children's inclusion in focus group discussions. In line with Bessell's advice on the design of children-centred social research, the researcher sought to cast the research appropriately for the children at whom it was aimed (Bessell 2006; cited in Redmond 2008, p. 10). The focus groups included children aged 10–14 years, mostly from class four and five, as they were deemed by their parents to be sufficiently mature to understand the theme and objectives of the research. Children's consent was sought verbally, and they were carefully observed for signs of explicit or de facto withdrawal of consent to ensure their rights as voluntary participants were respected. To build rapport and trust, the principal author conversed informally with children while playing local outdoors games such as Marbles, a game with small glass balls, and Gilli-Danda, a popular local game that is played with round stick of the size of baseball bat. Once the children were comfortable with the researcher and the research process, their views on the benefits of cash support in their families were elicited

through informal discussion and flip chart exercises. Their voices were recorded and then analyzed, first under the broad themes of material, relational and subjective well-being, and subsequently broken down under more specific themes of agency, education, livelihood, good life, satisfaction with life, future aspirations and wants through open coding method.

Material well-being: Against this determinant, a household survey was carried out to collect data to ascertain monthly household income, including the cash support given by the government through BISP and PBM. The survey also explored household expenditure patterns, and expenditure incurred on children and their needs.

Relational well-being: For this determinant, parents and children were asked about their views on children's participation in the decision-making processes in the household. Children were asked about their scope of individual agency, about the perceived impacts of social protection programmes on their lives and what links they perceived between their own deprivations and broader socioeconomic inequalities.

Subjective well-being: This determinant focused on the views of both parents/carers and children regarding what they considered important for a good life, their aspirations for the future and their level of satisfaction with life.

## 4. Findings

### 4.1. Cash Transfers and Material Well-Being of Children

The fieldwork data indicate that the cash transfer programmes had some positive impacts on families' and children's material well-being but limited capacity to address specific vulnerabilities of children. The main positive impacts are an increase in spending on food consumption and a small increase in health spending on children in beneficiary households. Conditions associated with the cash payments have compelled families to send children to schools. Nonetheless, some children still work in farms and small shops to earn money to fulfil some of their own needs and wants (such as snacks and toys) which their families could not afford to buy. As in Bessell's (2009) research with children in Indonesia, 'snacks' in the context of Pakistan does not refer to additional and unnecessary food items, but to small meals bought from vendors to supplement the limited food available in the family. This suggests that the CTPs operating in Shaheed Benazirabad district have addressed basic needs to a degree but not sufficiently. Children's comments in our study provide further evidence to Cheema et al.'s (2016) principal finding in their country-wide evaluation of BISP that it improved food consumption but not child nutrition. Its impact on poverty reduction was insignificant when Pakistan's Cost of Basic Needs methodology was used, which includes clothing, education and shelter (Cheema et al. 2016, pp. 45–46). This broader assessment is confirmed by the socioeconomic profile of the beneficiary households who took part in our study in Shaheed Banazirabad. Of the surveyed households, 82% were wage workers in agriculture, 10% self-employed, 6% sharecroppers as agriculture workers and 2% were seasonal agriculture workers (Table 2).

**Table 2.** Socio- economic profile and monthly income of sample households.

| Main Source of Livelihood | % of Sample Households |
|---|---|
| Agricultural workers (wage worker, sharecroppers or seasonal workers) | 90 |
| Self-employed | 10 |
| **Monthly Income (Including Cash Transfer Support by the Government), Pakistani Rupees (PKR)** | **% of Sample Households** |
| 5000 or less | 26 |
| 5001–7000 | 42 |
| 7001–10,000 | 30 |
| 10,001 and above | 2 |

Table 2 shows that the beneficiary households had monthly incomes below the minimum wage of PKR 12,000 per month announced by the government in 2010. Among surveyed households, only 2 percent had a monthly income of PKR 10,000 and above, while 26 percent of households earned PKR 5000 or less. Based on Pakistan's Cost of Basic Needs (CBN) poverty line, which assumes an adult equivalent consumption expenditure of PKR 3,244 and a weight of 0.8 of this rate for individuals younger than 18 years old (Cheema et al. 2016, p. 89), an average family of 6.5 persons would need to have almost PKR 20,000 per month to be above the poverty line. The households in our study are considered ultra-poor, and the cash support has helped these households sustain their basic dietary needs but has made no real impact on their situation. This is demonstrated by their responses to the question how they spend the CTP component of their income. As much as 73 percent of household CTP income is spent on food and nutritional requirements of family members (Table 3). The CTPs offered some income security for poor households plagued by inconsistent income and depreciating wage work. As one participant put it, "our income is irregular through casual labour. In that scenario, the cash support we get doesn't seem of much value but is a cushioning to help us live" (Bano, beneficiary mother).

**Table 3.** Monthly cash support spending patterns in beneficiary households.

| Monthly Cash Support Payment Grants (PKR) | Number of Households | Food | Children's Education | Health | Other/ Miscellaneous |
|---|---|---|---|---|---|
| 1700 | 5 | 61% | 20% | 19% | 0% |
| 1800 | 16 | 70% | 12% | 16% | 3% |
| 1900 | 4 | 68% | 15% | 12% | 5% |
| 2100 | 14 | 73% | 12% | 11% | 4% |
| 2300 | 11 | 65% | 14% | 12% | 9% |

In addition to food, cash grants have enabled families to spend around 11–20 percent on health and on the education of children (Table 3). The low spending on children's education could be because families receiving cash transfers remain deeply impoverished and unable to afford educational spending. As Table 4 indicates, in 71 percent of the children aged 15 years or above in surveyed households were illiterate and have never attended school, and 44 percent in the 11–14 age group were illiterate. However, the new enrolments of children between 5–10 years are encouraging, with 94 percent enrolled. This is to a large extent due to conditional cash support of PKR 750 per quarter provided under the Waseela-e-Taleem (WeT) Programme encouraging the beneficiary households to register their children between the ages of 4 and 12 (BISP 2018). As one child commented, "we have been enrolled into schools and urged by our families to attend schools regularly. We will keep receiving cash support under BISP and PBM if we make sure we keep up with the attendance" (Rahimdad, beneficiary child). Children seemed to be keenly aware of the need to juggle a variety of income sources for their family to survive.

As mentioned earlier, one of the most significant impacts of CTPs on children's material well-being is through school enrolment and associated nutritional and health benefits. However, this needs to be placed in context as the level of literacy among children is related to the education of their parents (Sabates-Wheeler et al. 2009). Among the surveyed households, an overwhelming majority of mothers assessed themselves as illiterate. While the literacy level reported by younger fathers was higher at 40%, the majority of fathers could not read and write (Table 4). The parents' lack of education and their poverty contributes to a passive stance towards their children's education. Many failed to see how education would translate into improved livelihoods. According to one informant, "studies haven't had positive outcomes for most of the youth in our area; they have been unable to finds jobs and contribute towards household incomes" (Fahmida, beneficiary mother). Despite this, families are ensuring children are enrolled into schools due to the conditions attached to their cash support.

**Table 4.** Parental literacy and child enrolments.

| Children's Age | % Enrolled/Literate | % Illiterate |
|---|---|---|
| 5–10 years | 94 | 6 |
| 11–14 years | 44 | 44 |
| 15 years and above | 8 | 71 |
| **Mothers Age** | **% Literate** | |
| 25–35 years | 15 | 85 |
| 36–45 years | 4 | 96 |
| 46–70 years | 11 | 89 |
| **Fathers Age** | **% Literate** | |
| 25–35 years | 40 | 60 |
| 36–45 years | 20 | 80 |
| 46–70 years | 13 | 87 |

Note: Figures do not add up to 100% due to missing data.

Interviews with mothers and discussions with children indicate that the CT payments conditional on school attendance have reduced child labour, but some children stated that they continued to work owing to their families' inability to take care of their needs and wants. Children revealed that they worked on farms and in shops to earn money, which they spent on food consumption and sometimes on toys. Their families were not financially able to provide more than basic food and clothing: "Parents don't usually give me snacks because they can't afford spending on such things; therefore I have to work either on the farm or in the motorcycle repair workshop to earn a few bucks for my own expenses" (Sabir Ali, beneficiary child).

Cash transfer programmes brought partial satisfaction of material needs among the children in our study. They perceived that their families had increased spending on food and health, and some believed their families bought them clothes and shoes more often compared to the times before cash transfer programmes started. As one child commented, "clothes we get more often, previously we used to get only two times a year" (Hashmat, beneficiary child). Children perceived that more was spent on food: "Since our family receives regular cash support, they spent more to buy food items" (Rahimdad, beneficiary child).

*4.2. Cash Transfers: Relational and Subjective Well-Being of Children*

Cash transfer programmes have not lived up to children's conceptions of a good life, which revolve around an economic self-sufficiency that would provide material goods they see with others around them, including good quality clothes and shoes and playgrounds with proper facilities: "One can buy everything with money including luxuries. This is what we see and witness in our surroundings; those who are financially strong live a good life" (Naghma, beneficiary child). This suggests that children relate the deprivation they experience to the economic incapacity and deep poverty of their families, which sets them apart from others in the community. While CTPs have alleviated some material constraints they have failed to significantly impact on social and psychosocial constraints, including the shame associated with poor quality clothing and exclusion from enjoyable playground facilities.

Although children do exercise agency in practical ways, by earning small amounts of money or helping their parents at home or on the farm, they are not considered capable of contributing to decision making in the family. Obedience to elders is considered a norm in Pakistani society to which everyone must adhere. Parents in our study unanimously held the view that children did not know what was good for them and what was not. They saw them as inexperienced minors, as one informant explained: "They don't understand what the complexities of our society are, therefore, until they achieve adulthood and get married, we take all the decisions" (Rubina, beneficiary mother). If children wanted to exercise strategic agency, for example, by insisting on being educated against their parents'

wishes, prevailing norms became stumbling blocks for them. As one child explained, "our parents take all the decisions regarding what we need, and what could be our needs; and similarly regarding any other family decision, including sending us to schools. I simply follow what they say; for instance, when I wanted to attend school, they said their financial situation didn't allow them to send me to school" (Daim Ali, beneficiary child).

Notions of relational and subjective well-being for both children and their parents are immersed in the reality of the prevailing socioeconomic class structures that are perceived as unchangeable. Among the adult respondents, for whom little has changed in their lifetime, there was a pervasive sense that life will remain the same, and any notion of a good life is tantamount to daydreaming:

> We were born with these circumstances, so are our children and so will be their children, these circumstances never change for us. Therefore, it is fair to say that communities like ours who have remained poor will remain poor and depend on alms from government or charitable organizations.
>
> (Rubina, beneficiary mother)

However, the lack of aspiration and hope in such comments must be placed in the broader context of a lacking commitment by the government to tackling systemic inequities and providing the means to fulfil basic human rights. As perceived by this parent, CTPs do not enable poor families to fulfil their aspirations for a better life:

> Mere small contributions every month or every quarter are not going to change our circumstances. We need to have access to jobs, skill enhancement, universal free education for our children; universal health care for our families, only then we can turn around our communities.
>
> (Parri, beneficiary mother)

Children shared the view that the cash payments were too low to make a real impact on their poverty. In response to the question how the CTP had impacted their lives, focus group participants offered the following realistic assessment:

> The amount our families are receiving is insufficient to meet the needs of whole households, specially the poverty we live with. Our families can hardly buy food with that, therefore, expecting these programmes to change our lives altogether would be erroneous.
>
> (Ghulam Abbas, beneficiary child).

> The support coming through these programmes is good, and our families understand that governments have realized our needs and circumstance we live in and the situation of job market. But if you think they are solution to all our problems, it would not be correct, we still work, our all needs are not fulfilled, we can't ask for what we want, we can't enjoy our lives as children of well-off families do (Waqar, beneficiary child).

These responses demonstrate that children held strong views about the nature and causes of the poverty they were experiencing and were acutely aware of their parents' constant worries about fulfilling the family's basic needs. Children acknowledged the cash transfers as a positive part of their households' repertoire of coping with extreme poverty. Furthermore, they took the programme as a sign that their families had not been totally forgotten by the country's decision makers. However, they clearly perceived the limitations of the CTPs in enabling families to even fulfil their basic nutrition needs, confirming Cheema et al.'s (2016, p. 50) assessment that the BISP failed to have a measurable impact on the country's child nutrition crisis. While demonstrating that poor families are not forgotten by the government, the programmes did not deliver enough to demonstrate that they are properly cared for. As both adult and child participants make clear, transformational change is well beyond the scope of Pakistan's CTPs, and so is enjoyment of childhood.

## 5. Discussion and Conclusions

The empirical research reported in this paper shows that CTPs in Pakistan aimed at improving the lives of poor families have had some modest success in that they encouraged beneficiary households to spend more on food and health needs. These benefits focus on material well-being and are informed by notions of relieving food and income insecurity that underpin the design of these programmes. The qualitative data reveals that while the cash payments are welcomed by parents and children as a component in their strategy of coping with extreme poverty, their size is insufficient to make a real improvement in the families' ability to meet their material needs and lifting them above the extreme poverty line. This finding is supported by impact evaluations of BISP, one of two Pakistani CTP examined here, which show that their impact on poverty is patchy when the UN's Multidimensional Poverty Index is used as a framework (Cheema et al. 2016). In addition to food consumption expenditure, which forms part of the health dimension of the MPI, it also measures living standards and education. Cheema et al. (2016) found increased expenditure on food among BISP beneficiaries and improvements in certain indicators related to living standards, particularly flooring, cooking fuel and assets, but in the indicators related to children, only school enrolment had improved, and this only after additional cash transfer payments were tied to school attendance. Even though many parents are illiterate, they recognize that education may be a way for their children to have a better life, and some make significant sacrifices to send their children to school (Cheema et al. 2016, pp. 67–68). However, most CTP recipients are too poor to do so without additional financial help, and some parents are skeptical about the transformative potential of primary education alone. Child participants in our study were aware of the links between their school attendance and conditional cash payments and felt responsible for ensuring the continued flow of these payments. While our interview data suggests that conditionality assisted school enrolment and attendance, this is mainly due to a combination of extreme poverty and high cost of education that puts education out of reach for poor families in the first place. The Pakistani government's investment in education was less than 3 percent of the GDP over the past decade (World Bank n.d.).

As Pakistan's CTPs were designed to provide a minimum income support package for the poorest households, our findings confirm other research that CTPs have at best delivered a modest improvement among poor families. However, they fall short of a strategy to pull these families out of the poverty trap. Children have been only indirectly considered as a target group for measuring nutrition, health outcomes and school enrolment. The vulnerabilities of children identified by Sabates-Wheeler and Roelen (2011) as distinct from those of adults have not been accounted for: their dependence on adults for their basic needs, their vulnerability to violence, abuse and social exclusion and their subordination to elders and invisibility in the public sphere. Excluded from decision making in their families and their broader community, children rely on adults to represent their needs and interests. In our study, parents strongly held the view that their children did not understand the complexities of life and should therefore obey their elders. While children had little option but to comply, they demonstrated a sophisticated understanding of their family's poverty as rooted within the broader socioeconomic and political context. Aware of the paramount importance of money to achieve a measure of independence and freedom, some children sought to earn income of their own to buy food or small items for their enjoyment. They shared similar views to their parents about the limited impact of CTPs but added important perspectives on the inability of CTPs to address their social exclusion and meet their psychosocial needs, such as feeling secure, access to playgrounds, and enjoying childhood.

### 5.1. Three Dimensional Well-Being and Limitations of the Study

The three-dimensional model of child well-being developed by Sumner (2010) proved useful in assessing the impacts of CTPs on children's well-being beyond the usual nutrition, health and education indicators. This research attempted to measure all aspects of material

well-being and despite its much more limited scale, it found remarkable parallels with the official evaluation of the BISP. However, the impact of the programmes on children's relational and subjective well-being is more difficult to measure, given the short span of cash transfer programmes and the persistence of socioeconomic conditions in which children and their families have been living for generations. Qualitative research methods are indispensable in trying to gain insights into children's views about their relational well-being, including their agency and level of participation in decision making. The focus groups with children revealed that they were aware of social exclusion as a result of poverty and of how their lives were deprived in comparison with better-off children. However, they seemed to have little awareness of their individual rights as children and did not mention any care networks beyond their own family. This suggests that social protection as currently conceptualized through CTPs is far from being a comprehensive policy which could lead to achieving children's well-being. To address extreme poverty in a child-sensitive manner, interventions are needed that provide economic security for care givers and take a rights-based approach that acknowledges children as current and future citizens.

Some limitations of this study should be noted. Firstly, it has not been able to probe the CTPs' impact on relational aspects of children's well-being that are highly sensitive, such as neglect, violence and abuse. This would require a more in-depth and longitudinal research approach that is backed up by tangible resources and measures to address vulnerabilities that might be revealed. Secondly, some subjective aspects of well-being related to children's concept of self, fears, aspirations, sense of meaning and trust were not investigated as they were considered to be too far removed from the aims and objectives of Pakistan's CTPs. Further research should focus on these well-being aspects and on the longer-term impacts of CTPs.

*5.2. Policy Recommendations*

While it could be argued that adopting a child-sensitive approach to social protection in Pakistan is an overly ambitious call, it is important to remember that children constitute a significant proportion of Pakistan's population. Ensuring their well-being is pivotal to a better future for the country as it is tackling the longer-term impacts of the COVID pandemic. A holistic approach linking the vulnerabilities and well-being of children to their rights can be a good way forward to address Pakistan's persistent poverty problems (UNDP 2016), also considering that the state should fulfil its obligation under the UNCRC to provide citizenship-based entitlements to children. As mainstream development policy planning is moving towards more transformative models of social protection, there are pressures for CTPs to go beyond basic income support (Molyneux et al. 2016). Incorporating specific initiatives aimed at addressing children's strategic needs (Roelen and Sabates-Wheeler 2012) and involving children in the design and evaluation of these programmes are necessary steps forward to improve children's well-being.

**Author Contributions:** Conceptualization, A.H.; Methodology, A.H. and S.S.; Formal Analysis, A.H. and S.S.; Investigation, A.H.; Writing—original draft preparation, A.H.; Writing—review and editing, S.S.; Supervision, S.S. All authors have read and agreed to the published version of the manuscript.

**Funding:** This research received no external funding.

**Institutional Review Board Statement:** Ethical review and approval were waived for this study, due to it being a required research component of a postgraduate degree that already had generic approval by the International Institute of Social Studies at the Erasmus University in Rotterdam.

**Informed Consent Statement:** Informed consent was obtained from all parties involved in this study.

**Data Availability Statement:** The data presented in this study are available on request from the corresponding author.

**Conflicts of Interest:** The authors declare no conflict of interest.

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
