# Peer review of "Cash Transfer Programmes in Pakistan through a Child Well-Being Lens"

_socsci, doi:10.3390/socsci10090330_

Round 1

Reviewer 1 Report

Referee report on “Cash transfer programmes in Pakistan through a child well-being lens” submitted to Social Sciences for publication under the code socsci-1291699.

Brief summary:

The paper examines the effect of two cash transfer programmes (CTPs), including Pakistan Baitul-Mal (PBM) and Benazir Income Support Programme (BISP), on child well-being in terms of material needs, relational vulnerabilities and subjective views of well-being in Pakistan. The paper used a survey of fifty household, which was undertaken in Shaheed Benazirabad district of Sindh province where both programmes were rolled out and was conducted in July and August 2015, to explore the relationship between CTPs and child well-being. Based on some descriptive statistics, the authors concluded that CTPs had some positive impacts on families’ and children’s material well-being in terms of an increase in food consumption expenditure, whereas CTPs had limited capacity to address other non-material dimensions of child well-being.

Main comments:

I think the potential contribution of the paper may lie in an attempt to combine several quantitative and qualitative data sets, including a household survey, interviews with mothers in beneficiary families and group discussions with children, to understand the effect of CTPs on child well-being. However, although the authors stated that “The empirical research reported in this paper shows that CTPs in Pakistan aimed at improving the lives of poor families have had some modest success in that they encouraged beneficiary households to spend more on food and health needs” on page 10, there was not any empirical analysis conducted throughout the whole paper. Indeed, the paper just presented some descriptive statistics and obtained findings. As a result, I had trouble interpreting the effect of CTPs on household consumption expenditure shown in Table 3 on page 8. I wonder why the authors could reach a conclusion that “The main positive impacts are an increase in spending on food consumption and a small increase in health spending on children in beneficiary households (page 7)” when all of the households were participated in both programmes and health spending seems to decrease with monthly cash support payment grants.

I suggest the authors should use their survey data to conduct some regression analyses. Because their data set was at the household level, the authors could first run several regressions on rate of primary school enrollment, rate of children’s illiterate and consumption expenditure. The authors could consider including number of beneficiary children (continuous or categorical variable), monthly family earning and parents’ age and education as control variables. The effect of CTPs could then be identified by the coefficient of the number of beneficiary children variable. I think it would be clearer to interpret the effect of CTPs on child well-being in terms of education and material needs based on empirical models. In addition, I suggest the authors could supplement empirical findings with their qualitative data sets, which were obtained by interviews and group discussions.

Minor comments:

  1. The authors stated that “While BISP has been extensively evaluated (Cheema et al, 2016), there has to date been little attention paid to children’s well-being (page 6).” I suggest the authors should briefly review relevant literature on BISP and, then, compare differences in data, methodology and findings between prior studies and this paper.
  2. I suggest the authors could integrate Tables 2-5 into one or two tables to summarize the descriptive statistics of the survey data. In addition, the authors could attempt to present their qualitative findings under the categories of child well-being shown in Table 1.
  3. I suggest the authors should provide the survey questionnaire and the topics of interviews and group discussions as appendices.

Reviewer 2 Report

Thank you for the opportunity to review this paper.

I do have some comments:

The introduction is well written. The project scope and issue that the research addresses are clearly defined.

Methods

Households were identified through purposive sampling. Perhaps need to state that these households are representative of the Pakistan Baitul-Mal and Benazir Income Support Programme beneficiary households or otherwise? Would also have been helpful to know the proportion of households in each parent age group and also household size.

It is unclear if the survey, semi-structured interviews and group discussions with children designed a priori. It is also unclear what the results of the survey were cross-tabulated with to generate information about household well-being.

Findings

Structuring findings as per methods section with subheadings might improve readability.

Discussion and conclusion

The discussion is nicely summarised for readers, with context to existing literature. 

Reviewer 3 Report

See attached.

Round 2

Reviewer 1 Report

Thank you for your careful responses. The revised paper is fine. However, because this is a qualitative study, I still hope that the authors can supplement the questionnaire and part of main qualitatve responses as appendices if possible.

Author Response

Thank you very much for engaging with our work and providing advice to improve the paper. Please see our replies below to your points.

Comment-1, Thank you for your careful responses. The revised paper is fine. However, because this is a qualitative study, I still hope that the authors can supplement the questionnaire and part of main qualitative responses as appendices if possible.

Response-1. We will upload the survey questionnaire and the interview questionnaire as supplementary information.

Reviewer 3 Report

Almost all the comments were responded to appropriately. I disagree with one, on benchmarking to other countries being inappropriate. It is not inappropriate, it would actually be useful in making claims about the size of a transfer being small. In any case, this is a minor point and does not require revision. 

Author Response

Thank you very much for engaging with our work and providing advice to improve the paper. Please see our replies below to your points.

Comment-1 Almost all the comments were responded to appropriately. I disagree with one, on benchmarking to other countries being inappropriate. It is not inappropriate, it would actually be useful in making claims about the size of a transfer being small. In any case, this is a minor point and does not require revision.

Response-1, We appreciate your suggestion and will bear it in mind for further research. However, in this paper we believe it is more relevant to keep the benchmarking and frequency of the transfer focused on Pakistan. On p. 9 we have referenced Cheema et al’s (2016) evaluation of the BISP program that concludes the transfers are too small to have an impact on the poverty line as calculated by the government of Pakistan. On p. 6 we included the equivalent sum in US$.